# Experimental Study on Toughness of Engineered Cementitious Composites with Desert Sand

**DOI:** 10.3390/ma16020697

**Published:** 2023-01-11

**Authors:** Zhishuan Lv, Yang Han, Guoqi Han, Xueyu Ge, Hao Wang

**Affiliations:** 1College of Civil Engineering, Kashi University, Kashi 844008, China; 2Institute of Engineering Disaster Prevention and Mitigation, Henan University of Technology, Zhengzhou 450001, China

**Keywords:** desert sand, engineered cementitious composites (ECCs), particle size, uniaxial tension, toughness

## Abstract

In this paper, engineered cementitious composites (ECCs) were prepared with desert sand instead of ordinary sand, and the toughness properties of the ECCs were studied. The particle size of the desert sand was 0.075–0.3 mm, which is defined as ultrafine sand. The ordinary sand was sieved into one control group with a size of 0.075–0.3 mm and three other reference groups. Together with the desert sand group, a total of five groups of ECC specimens were created. Through a uniaxial tensile test, three-point bending test and single-seam tensile test on the ECC specimens, the influence of aggregate particle size and sand type on the ECC tensile strength, deformation capacity, initial crack strength, cement-matrix-fracture toughness, multiple cracking characteristics and strain-hardening properties were studied. The experimental results show that the 28d tensile strain of the four groups of the ordinary sand specimens was 8.13%, 4.37%, 4.51% and 4.23%, respectively, which exceeded 2% and satisfied the requirements for the minimum strain of the ECCs. It is easier to achieve the ECC strain hardening with sand with a fine particle size; thus, a particle size below 0.3 mm is preferred when preparing the ECCs to achieve a high toughness. The multiple cracking performance (MCP) and the pseudostrain hardening (PSH) of desert sand and ordinary sand with a 0.075–0.3 mm grain size were 2.88 and 2.33, and 8.76 and 8.17, respectively, all of which meet the strength criteria and energy criteria and have similar properties. The tensile strength and tensile deformation of the desert sand group were 4.97 MPa and 6.78%, respectively, and the deformation capacity and strain–strengthening performance were outstanding. It is verified that it is feasible to use desert sand instead of ordinary sand to prepare the ECCs.

## 1. Introduction

Since its first appearance in the middle of the 18th century, concrete has been widely used in various industrial and civil buildings, roads, bridges and other infrastructures due to its high strength, extensive sources of raw materials, strong applicability and low cost, in addition to other advantages. However, because of its low tensile strength, poor ductility, easy-cracking characteristics and deterioration under environmental effects, the problem of concrete durability is prominent, which increases the cost of maintenance and protection and restricts the development of concrete materials [1,2]. Almost all infrastructures that are damaged due to a lack of toughness, durability and sustainability can be traced back to tensile cracking and the fracture of concrete.

To overcome the defects of traditional concrete materials, such as high energy consumption, a high degree of brittleness failure and a poor crack-control ability, engineered cementitious composites (ECCs) have been developed and the strength criteria and energy criteria based on micromechanics were proposed by Victor C. Li. A theoretical basis based on these two criteria was provided for the engineering application of the ECCs [3,4,5]. If both the strength criteria and the energy criteria are satisfied, the characteristics of the multiple cracking and strain hardening of the ECCs under tension can be realized. The uniaxial tensile strain of the ECCs can exceed 2% [6,7,8], which is more than 200 times that of ordinary cement-based materials [9]. In the failure process for the ECCs, many fine cracks are produced with a crack width that is generally within 100 μm [10,11,12], and the invasion of harmful substances can effectively be prevented and the impermeability [13,14,15], self-healing ability [16,17,18] and durability [19,20,21,22] of concrete can be improved. The research on the ECCs from different perspectives and in combination with different factors has been conducted by scholars all over the world [23,24,25,26].

A large number of industrial wastes are used in the ECC preparation process to replace some or all of the cement, such as fly ash, slag, lithium slag and red-mud slag [27,28,29,30,31], and the energy consumption of the material-production process is greatly reduced, which conforms to the goal of achieving environmental sustainability [32]. Aggregates are an important component of the ECCs, which account for a large volume proportion and significantly affect the workability, strength, elastic modulus, ductility and other properties of the ECCs. In addition, aggregates can also reduce the production costs of the ECCs. At present, most ECCs are made of micro-silica sand from river sand, which is a nonrenewable resource. Because of the surge in demand for building raw materials, China uses approximately 20 billion tons of sand and stone every year, which account for half of the world’s consumption. The phenomenon of indiscriminate excavation and mining frequently occurs, which has caused great damage to the environment and has led to a sharp increase in the price of materials, such as construction sand, and a decrease in sand reserves [33]. Today, ECCs cannot be widely used in practical projects due to their high cost, and the existing research on reducing ECC costs has focused on the optimization and selection of fibers. However, the micro-silica sand used in the ECCs is also one of the important reasons for the high cost. Thus, finding new and alternative sand sources is important and urgent.

Desert sand is a very rich natural resource, which is widely distributed all over the world. The total desert area in China is approximately 700,000 km^2^, which accounts for 7% of its total land area. China has eight deserts with huge reserves of desert sand [34]. Desert sand is ultrafine sand with an average particle size generally below 0.2 mm. Currently, it is used in some concrete materials [35,36,37,38]; however, the use of desert sand in ECCs is rarely reported. If desert sand can be reasonably used in ECC materials, it will not only reduce engineering costs, but also protect environmental resources and help achieve the sustainable development of concrete.

Based on the above research contents, firstly, the chemical composition of the desert sand is tested to determine whether it contains harmful elements that can affect ECCs. Secondly, ECCs are prepared with desert sand, which are then compared with the ECCs of ordinary sand with different particle sizes. Through a uniaxial tensile test, three-point bending test and single-seam tensile test on the ECC specimens, the influences of desert sand and ordinary sand with different grain sizes on the ECC tensile strength, deformation capacity, initial crack strength, cement-matrix-fracture toughness, multiple cracking characteristics and strain-hardening properties were studied. The schematic flow diagram of this study is shown in Figure 1.

## 2. Materials and Methods

### 2.1. Materials and Preparation of ECCs

The cement used in this study was P·II 42.5 ordinary Portland cement produced by Xinjiang Tianshan Cement Co., Ltd. (Urumqi, China), and the mineral powder was high-quality first-grade S95. The fine aggregates were desert sand from Wuqia County, Kezhou and ordinary river sand from Kashi. The fiber used in this study was polyethylene (PE) fiber. The polycarboxylic acid superplasticizer was produced by the Kashi Water Reducing Agent Factory, as was the 200,000-viscosity hydroxypropyl methyl cellulose. Group A was composed of desert sand with a particle size of 0.075–0.3 mm. The ordinary river sand was sieved into four particle size grades with particle size ranges from 0.075 to 0.3 mm for Group B, from 0.3 to 0.6 mm for Group C, from 0.6 to 1.18 mm for Group D and from 0.075 to 1.18 mm for Group E. The five groups of sands with different types and particle sizes were taken as the research objects, as shown in Figure 2. The particle size of the desert sand was very close to that of the ordinary sand, with a particle size from 0.075 to 0.3 mm. The chemical composition of the desert sand is listed in Table 1. The sulfide and sulfate contents were relatively low (calculated according to the mass of SO_3_) and the chloride content was very low (calculated using the mass of chloride ion), which satisfy the limit for harmful substances in the project. Table 2 lists the physical and mechanical properties of the PE fiber.

The five groups of ECCs were configured according to the five groups of sand and adopted the same mix proportion, as shown in Table 3. The success of the ECC preparation is closely related to the manufacturing process [39,40,41], especially regarding the mixing sequence and mixing time of materials. Cement, slag, sand and other materials were added to a 5 L planetary mixer according to the proportions listed in Table 3. After the materials were slowly mixed for 2 min, one-third of the water was added and stirred for 1 min. Then, the water reducer and one-third of the water were uniformly mixed and poured into the mixer. Subsequently, the remaining one-third of the water was poured into the mixed materials for 2 min of mixing. At this time, the fluidity of the cement matrix was very good. Then, the thickener was added for quick mixing for 1.5 min. Here, the cement matrix was relatively thick and dough-like. Finally, the PE fiber was uniformly dispersed into the cement matrix within 6 min. No obvious agglomeration occurred when the mixture was manually kneaded, which is a key indicator of the success of the ECC preparation. The final mixing time depends on the even dispersion of the fiber in the cement matrix without agglomeration. The prepared ECC material was put into a specially made transparent acrylic template and was subjected to full vibrations until large air holes no longer appeared, which was observed from the side and bottom of the template. The templates were removed after the prepared samples were kept for 24 h. The specimens were placed in a constant-temperature and humidity-curing room at a temperature of 20 ± 1 °C and a relative humidity of 95% for curing.

### 2.2. Test-Scheme Design

#### 2.2.1. Uniaxial Tensile Test

The main methods for testing the tensile properties of materials include uniaxial tensile, splitting tensile and bending tests. Among them, the most direct test method that can best reflect the tensile properties of the materials is the uniaxial tensile test. Although researchers in various countries have been trying to agree on the standardization of the uniaxial tensile test, it has not yet been standardized. At present, dumbbell-shaped specimens and plates are widely used in uniaxial tensile tests, and dumbbell-shaped specimens were used in this study [42]. Both ends of the dumbbell-shaped test piece consist of clamping ends. The gauge length of the test piece is 100 mm, and the size of the test section is 100 mm × 30 mm × 15 mm, as shown in Figure 3. A set of displacement sensors were installed at both ends of the test piece with a measurement range from 0 to 20 mm, and the final deformation was measured according to the average value of two sets of displacement sensors, as shown in Figure 4. An electronic tensile-testing machine with a measurement range of 5 kN, which was manufactured by Jinan Chuanbai Instrument and Equipment Co., Ltd. (Jinan, China), was used as the loading equipment, and displacement-control loading was adopted at a loading rate of 0.5 mm/min. The test pieces were divided into five groups according to the aggregate size and sand type. The curing ages were 7 and 28 days. Three test blocks were made at each curing age.

#### 2.2.2. Three-Point Bending Test

To quantitatively analyze the strain–strengthening and toughness properties of the desert sand and ordinary sand with different grain sizes, the fracture energy of the cement matrix needs to be determined according to the strength criteria and energy criteria [3]. When the three-point bending beam is a standard specimen (the span–height ratio of the specimen is four), the fracture energy of the ECC cement matrix can be tested according to the formula recommended by Tada [42]. A cement matrix specimen without PE fiber was made according to the mixing proportions listed in Table 3. The specimen size was 350 mm × 75 mm × 40 mm. Each group contained three test pieces, for a total of five groups with fifteen test pieces. After the matrix was cured under standard conditions for 28 days, the three-point bending test was performed at a loading rate of 0.5 mm/min. The test-piece span was 300 mm, and an incision that was 30 mm deep and less than 1 mm wide was made at the bottom of the test piece. The sizes of the test piece and test device are shown in Figure 5 and Figure 6, respectively.

#### 2.2.3. Single-Seam Tensile Test

To determine the parameters of the strength criteria and energy criteria, the maximum peak stress and maximum complementary energy should be determined based on the relationship between the bridging-fiber stress and crack width, in addition to the fracture energy of the cement matrix that is determined with the three-point bending test. To determine the relationship between the ECC stress and crack opening width, the multiple cracking needed to be artificially limited to ensure that single-seam cracking occurred in the case of failure. The size of the test piece was the same as that of the dumbbell-type test piece mentioned earlier. Each group contained three test pieces, for a total of five groups of fifteen test pieces. After 28 days of curing under standard conditions, an annular notch with a width of less than 1 mm was cut into the middle of the test piece. The depth of the notch is shown in Figure 7. The notch is cut using diamond-cutting pieces. During the notch-making process, care is taken to avoid damage to the other parts of the test piece to ensure that the ECC only experiences single-seam cracking when under uniaxial tension. The tensile test device is shown in Figure 8.

### 2.3. Evaluation Method of ECC Toughness

On the basis of the design theory of ECC micromechanics [3], the ECCs must satisfy both the strength criteria and energy criteria to achieve the characteristics of multiple cracking and strain hardening; otherwise, stress softening occurs during the tensile process.

(1) Strength Criteria

The strength criteria set the boundary condition of tensile stress for cracks starting from the initial defect, as they control the cracking process. Continuous multiple cracking must satisfy Equation (1).
(1)$$\sigma_{c}<\min\{\sigma_{0}\}$$
where *σ*_c_ is the initial crack strength of the matrix and *σ*_0_ is the peak stress.

(2) Energy Criteria

The crack-distribution phenomenon of the test conforms to the flat crack-propagation mode and satisfies the energy criteria. The stability of the crack width under a constant external load must satisfy Equation (2).
(2)$$\sigma_{0}\delta_{0}—\int_{0}^{\delta_{0}}\sigma (\delta )d\delta =J_{b}^{\prime }\geq J_{\mathrm{tip}}$$
where *δ*_0_ is the displacement corresponding to the bridging-fiber peak stress, *σ*(*δ*) denotes the relationship between the bridging-fiber stress and crack opening width, *J*′_b_ is the maximum complementary energy and *J*_tip_ is the fracture energy of the matrix material.

Theoretically, *σ*_0_/*σ*_c_ ≥ 1 and *J*′_b_/*J*_tip_ ≥ 1 can achieve stable tensile strain-hardening characteristics. However, in practice, because of the existence of uncertainty factors such as material fluctuation, an uneven manufacturing process, and test errors, satisfying the requirements for strain-hardening characteristics is difficult. Research shows that only PCM = *σ*_0_/*σ*_c_ ≥ 1.3 and PSH = *J*′_b_/*J*_tip_ ≥ 2.7 can meet the characteristics of multiple cracking and strain hardening [43], where MCP is the multiple cracking performance and PSH is the pseudostrain hardening.

Based on the calculations of Equations (3)–(6) recommended by Tada [41], which are combined with the three-point bending test, the fracture energy of the cement matrix can be calculated.
(3)$$J_{\mathrm{tip}}=\frac{K_{m}^{2}}{E_{m}}$$
(4)$$K_{m}=\frac{3(F+10^{-3}mg/2)10^{-3}L\sqrt{a}}{2bh^{2}}f\left(\alpha \right)$$
(5)$$f\left(\alpha \right)=\frac{1.99-\alpha \left(1-\alpha \right)\left(2.15-3.93\alpha +2.7\alpha^{2}\right)}{\left(1+2\alpha \right){\left(1-\alpha \right)}^{3/2}}$$
(6)$$\alpha =\frac{a}{h}$$
where *J*_tip_ (J/m^2^) is the fracture energy, *K*_m_ (MPa·m^1/2^) is the fracture toughness, *E*_m_ (GPa) is the tensile modulus of elasticity, *F* (kN) is the three-point bending peak load, *m* (kg) is the test-piece quality, *g* (m/s^2^) is the gravitational acceleration, *L* (m) is the span of the three-point bending test piece, *a* (m) is the notch depth, *b* (m) is the width of the test piece, *h* (m) is the height of the test piece and *f*(*α*) is the test-piece shape parameters.

According to the single-seam tensile test, the relationship between the stress and crack opening width can be obtained, and the maximum complementary energy can be obtained by integrating it with the axis where the stress is located, as shown in Figure 9.

## 3. Results and Discussion

### 3.1. Uniaxial Tensile Results and Stress–Strain Analysis

Figure 10 shows the failure mode of each group of specimens under uniaxial tension, and Figure 11 shows the stress–strain curve under uniaxial tension. The test phenomenon and stress–strain curve show that in the initial stage of loading, the ECC specimen is in the elastic stage and the tensile force is mainly borne by the cement matrix. When the cracking strength of the cement matrix is reached, the specimen exhibits the first crack, and the stress–strain curve suddenly drops. Cracks continue to appear as the tensile force increases and the stress–strain curve repeatedly fluctuates, exhibiting strain-hardening characteristics. During the failure process of the test piece, the continuous sound of fibers being pulled out or broken can be heard. The reason for this is that the PE fiber plays a bridging role after the cement matrix breaks, bears the load from the matrix and makes the surrounding matrix continuously generate new cracks until the PE fiber fails in its bridging role, resulting in multiple cracks and a large deformation.

The shape and distribution of the two groups of fractures are basically consistent based on the comparative analysis of Group A with desert sand and Group B with ordinary sand as both have a spacing of approximately 2–3 mm and a fracture width of less than 100 μm. The number of cracks is 40–50, which demonstrates the characteristics of the dense, saturated multiple cracks. The deformation of the two groups is mainly generated by the cumulative width of these multiple cracks. For ordinary sand, Group B exhibits dense, saturated multiple cracking characteristics. Although Groups C, D and E also exhibit multiple cracking characteristics, they exhibit unsaturated multiple cracking. The cracks are unevenly distributed, and the maximum spacing of the cracks can reach 15 mm, which indicates that the reinforcement effect of the fibers has not been fully utilized. The deformation of materials is mainly obtained through the final main-crack cracking, which limits the final deformation ability of the materials. In total, the number of cracks becomes increasingly fewer as the particle size increases. A coarse particle size is not conducive to the realization of multiple cracking in the ECCs, especially for particles larger than 0.6 mm. Desert sand and ordinary sand with a 0.075–0.3 mm particle size show excellent multiple cracking characteristics and the ability to control the crack width.

The uniaxial tensile properties of the ECCs with different particle sizes and ages are shown in Figure 12. From the perspective of the initial crack strength, the 7-day initial crack strengths of Group A with desert sand and Group B with ordinary sand are 2.09 MPa and 2.06 MPa, respectively, and the 28-day initial crack strengths are 2.51 MPa and 2.64 MPa, respectively, under the same change trend. For ordinary sand, the initial crack strength of the ECCs tends to increase along with the increase in the sand particle size, but the initial crack strength of Groups C–E at 28 days does not significantly increase, which may be caused by the random difference in the cement matrix’s defect size.

According to the 28-day tensile elastic modulus, for Groups A and B, the tensile elastic moduli are 3.76 GPa and 3.31 GPa, respectively. The tensile elastic modulus of the ECCs with desert sand is higher, but the difference is not significant. For ordinary sand, the tensile elastic modulus of Group B with fine particles is higher than the tensile elastic modulus of Groups C–E. The tensile elastic modulus decreases as the particle size increases.

Comparison and analysis of the desert and ordinary sand reveal that the tensile strengths of Group A at 7 and 28 days are 4.47 MPa and 4.97 MPa, respectively, and those of Group B are 4.06 MPa and 4.57 MPa, respectively. The tensile strength of the desert sand is slightly higher than that of Group B with ordinary sand. The tensile strains of Group A at 7 and 28 days are 5.80% and 6.78%, respectively, whereas those of Group B are 4.06% and 8.13%, respectively. The tensile strain of ordinary sand at 28 days is slightly higher than that of the desert sand, and the tensile deformation of both is much higher, being 2% more than the minimum limit value of the ECCs defined by the general rules, which indicates that the deformation capacity of the ECC materials prepared from desert sand is excellent and can completely replace ordinary sand.

The tensile strength and tensile deformation of ordinary sand with a 0.075–0.3 mm particle size show an increasing trend alongside the increase in age, whereas when the particle size exceeds 0.6 mm, a decreasing trend occurs. The reason for this is that the strain-hardening property of the coarse-grain-sized sand decreases as the grain size increases, which leads to the premature occurrence of the main cracks in the test piece and leads to a decrease in the tensile deformation and tensile strength. The tensile strain of the four groups of the ECC specimens with different particle sizes exceeds 2%, which is consistent with the identification of the ECCs in this study. Sand with a grain size of less than 0.3 mm can effectively improve the deformation capacity and tensile strength of the ECCs. The 0.3–0.6 mm grain size increases with age, and the change range of the tensile strength and tensile deformation is not obvious. A coarse-grain size of more than 0.6 mm can reduce the deformation capacity and tensile strength, which is not conducive to achieving strain strengthening in the ECC materials.

### 3.2. Three-Point Bending Test Results and Fracture Energy of the Cement Matrix

Figure 13 shows the failure mode of the three-point bending test piece. Because PE fiber was not added to the cement matrix, each group of the test pieces was brittle when they were damaged, and they all broke from the notch. Equations (3)–(6), which are recommended by Tada [42], show that, based on the peak load of the specimen failure, the mass of the specimen and the elastic modulus measured under uniaxial tension, the fracture energy of the five groups of matrixes can be obtained, as listed in Table 4 and Figure 14.

For Group A and Group B, the fracture energies are 72.5 J/m^2^ and 67 J/m^2^, respectively, which are small and conducive to achieving a high toughness. For ordinary sand, the matrix fracture energies of Groups B to E are 67.0 J/m^2^, 90.6 J/m^2^, 109.6 J/m^2^ and 96.5 J/m^2^, respectively. The matrix fracture energy of Group B was the smallest, and that of Group C, Group D and Group E was 35.2%, 63.6% and 44.0% higher than that of Group B. It can be seen that the larger the particle size is, the higher the fracture energy is.

### 3.3. Single-Seam Tensile Test Results and Complementary Energy

Figure 15 shows the failure mode of the single-seam tensile specimen. The failure occurs at the notch, and no cracks appear in other places. The stress–displacement curve is shown in Figure 16. The peak stress and the opening width at the peak stress can be obtained. The complementary energy *J*′_b_ can be obtained by integrating the axis of the stress, as listed in Table 5.

### 3.4. Analysis and Discussion of Fracture Toughness

The multiple cracking performance (MCP) and the pseudostrain hardening (PSH) could be obtained, respectively, according to the matrix’s fracture energy *J*_tip_, the complementary energy *J*′_b_ and the initial crack strength, which was measured by the three-point bending test, the single-seam tensile test and the uniaxial tensile test, respectively, as shown in Table 6 and Figure 17.

The MCP of Group A with desert sand and Group B with ordinary sand is 2.88 and 2.33, respectively. The desert sand group’s MCP is slightly larger than that of the ordinary sand group, and both are greater than 1.3, which meets the requirements of the strength criteria in Equation (1). Both groups of specimens have obvious characteristics of multiple cracking, which have been verified via the uniaxial tensile stress–strain curve and the specimen failure phenomenon. The PSH of the two groups of specimens is 8.76 and 8.17, respectively, which are both much larger than 2.7. The desert sand group’s PSH is slightly larger than that of the ordinary sand group, and both meet the requirements of the energy criteria in Equation (2). The higher PSH of the desert sand group is conducive to achieving a high toughness.

For ordinary sand, the MCP of Groups B to E is 2.33, 2.44, 2.45 and 2.40, respectively, which are all greater than 1.3 and meet the requirements of the strength criteria in Equation (1). The four groups of specimens have the characteristic of multiple cracking. The PSH of the four groups of ordinary sand specimens are 8.17, 5.39, 5.05 and 6.00, which are all greater than 2.7 and meet the requirements of the energy criteria in Equation (2). It can be seen that the smaller the grain size of the ordinary sand is, the easier it is to achieve stable multiple cracking and strain hardening. Similar conclusions were reflected in M. Sahmaran’s research [33]. In that research, dolomite limestone sand and gravel sand with a maximum particle size of 1.19 mm and 2.38 mm were used to replace micro-silica sand with maximum particle sizes of 0.2 mm when preparing the ECCs, and the production cost of the ECCs was reduced. The tensile strength and deformation of the ECC materials prepared by using larger grains of sand were reduced to varying degrees.

## 4. Conclusions

(1)The results show that the uniaxial tensile strength of the ECCs with desert sand is 4.97 MPa, whereas the maximum tensile strain is 6.17%. The high toughness of the ECCs with desert sand is shown to be outstanding according to the results of the stress–strain curves, the strength criteria and the energy criteria. It is verified that desert sand is feasible to be used as a substitute to ordinary sand for the ECC preparation.(2)The performance of the ECCs with ordinary sand is closely related to the particle size of sand. The tensile strength, tensile deformation and toughness of the ECC materials are decreased when the particle size is increased. The maximum tensile strength and tensile deformation were obtained for the particle size from 0.075 to 0.3 mm, which are 4.57 MPa and 8.13%, respectively. In engineering projects, the ECCs should be prepared with a particle size below 0.3 mm, and the maximum particle size should not exceed 0.6 mm.(3)The compression and bending properties of the ECCs with desert sand, as well as the interfacial connection between the fiber and matrix at the microscale, should be further studied.

## Figures and Tables

**Figure 1 materials-16-00697-f001:**
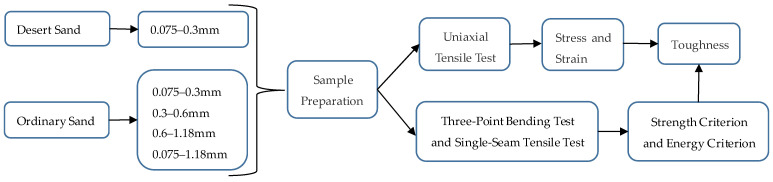
Schematic flow diagram of this study.

**Figure 2 materials-16-00697-f002:**
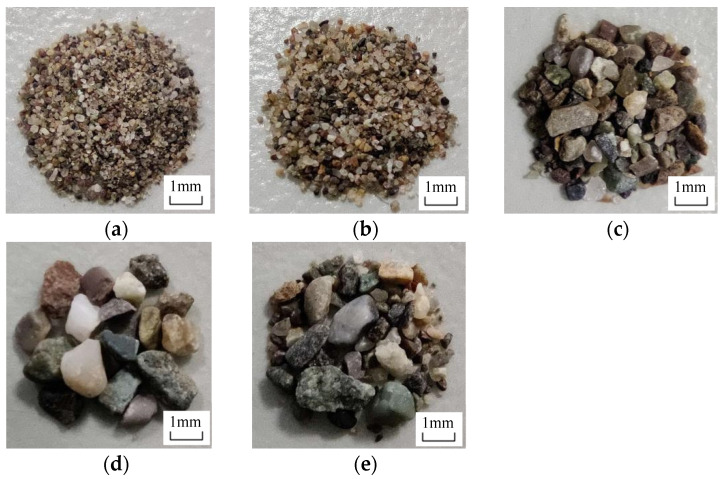
Desert sand and ordinary sand: (**a**) Group A, 0.075–0.3 mm; (**b**) Group B, 0.075–0.3 mm; (**c**) Group C, 0.3–0.6 mm; (**d**) Group D, 0.6–1.18 mm; (**e**) Group E, 0.075–1.18 mm.

**Figure 3 materials-16-00697-f003:**
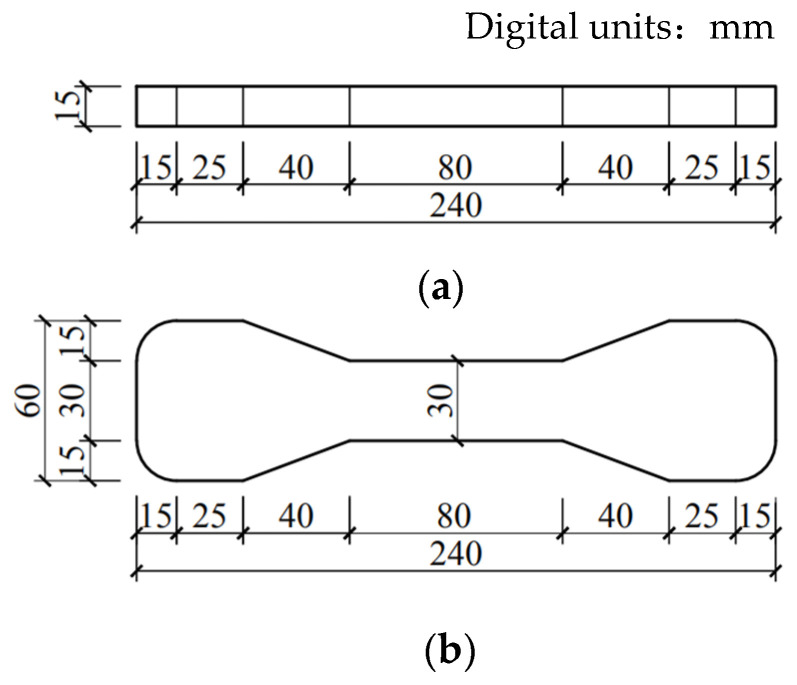
Schematic diagram of uniaxial tensile specimen: (**a**) profile; (**b**) plan.

**Figure 4 materials-16-00697-f004:**
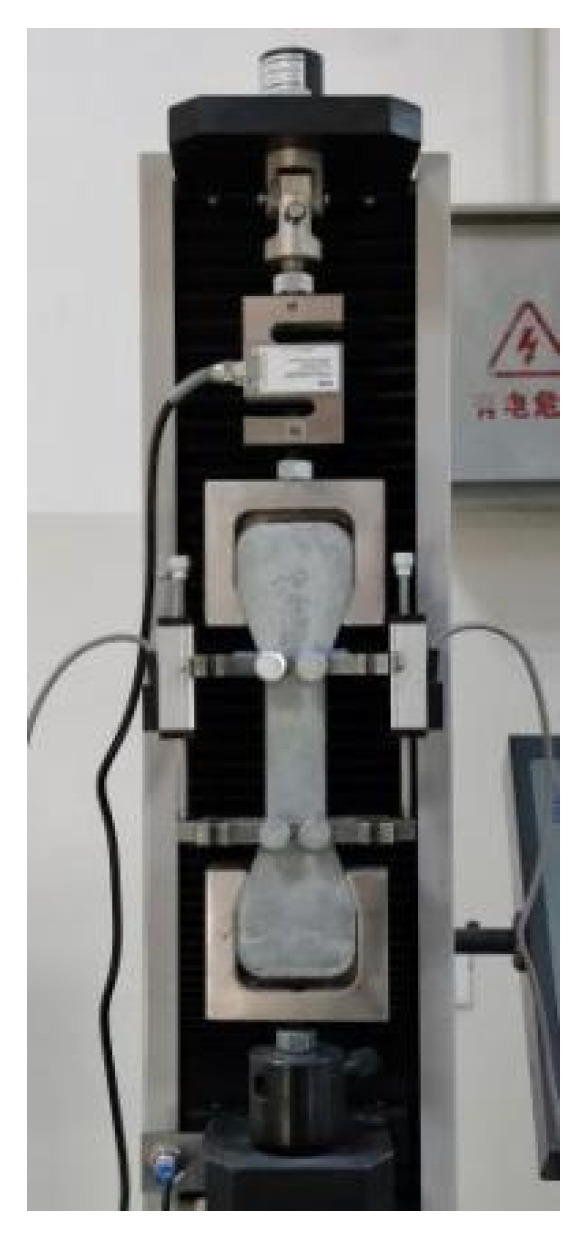
Uniaxial tensile test device.

**Figure 5 materials-16-00697-f005:**
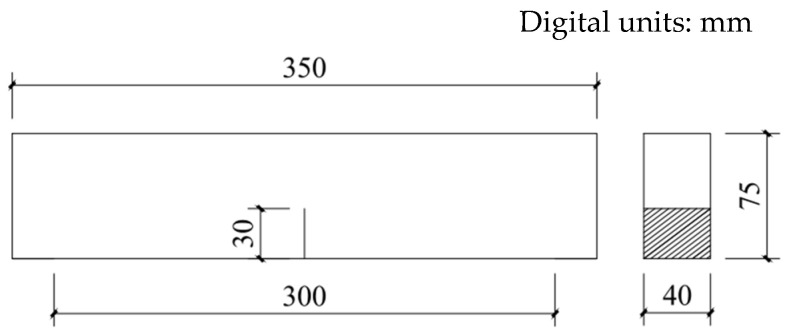
Dimensions of three-point bending test piece.

**Figure 6 materials-16-00697-f006:**
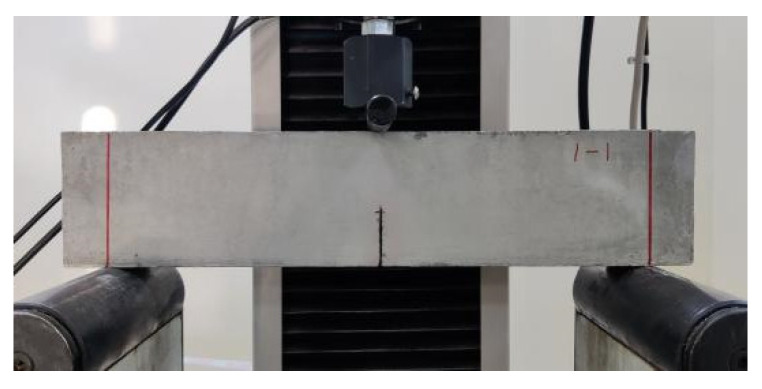
Three-point bending test loading device.

**Figure 7 materials-16-00697-f007:**
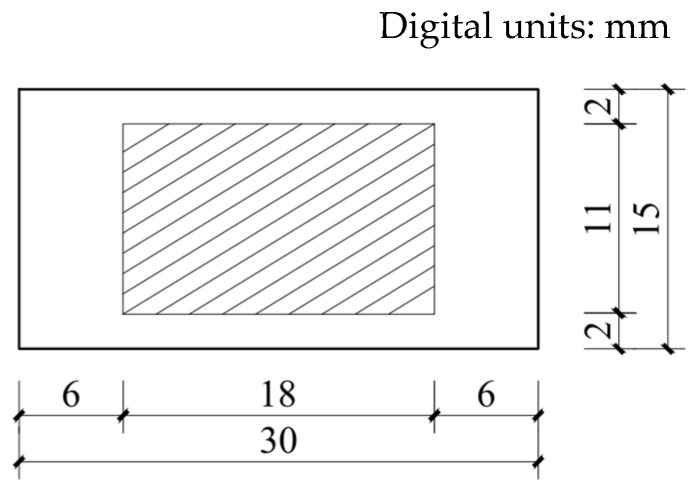
Cutting depth of the single-seam cracking specimen.

**Figure 8 materials-16-00697-f008:**
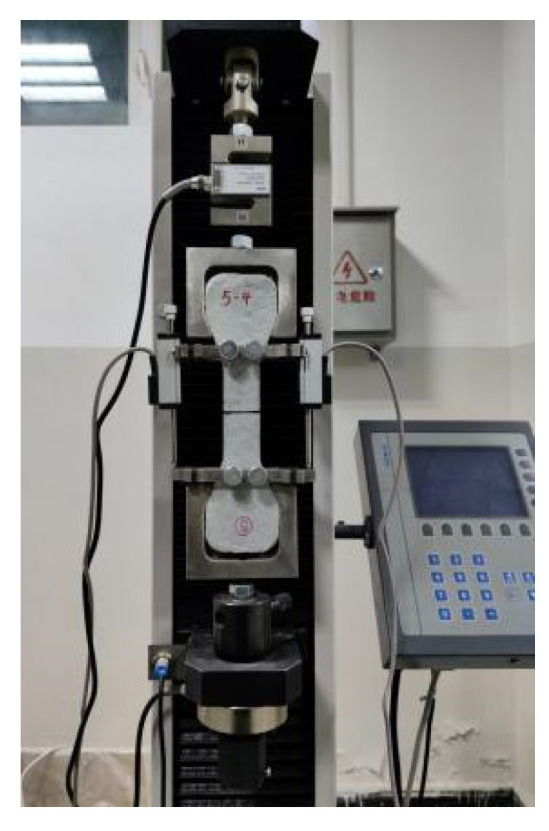
Single-seam cracking tensile test device.

**Figure 9 materials-16-00697-f009:**
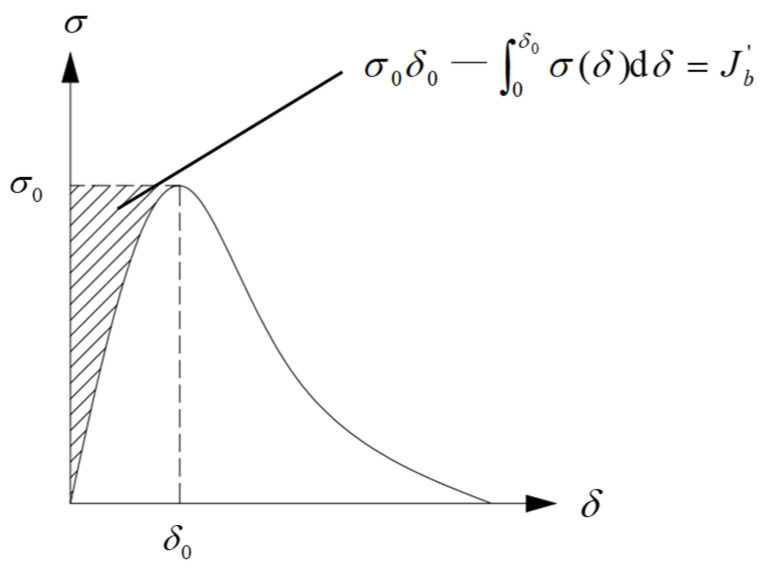
Relationship between the crack opening width and bridging-fiber stress.

**Figure 10 materials-16-00697-f010:**
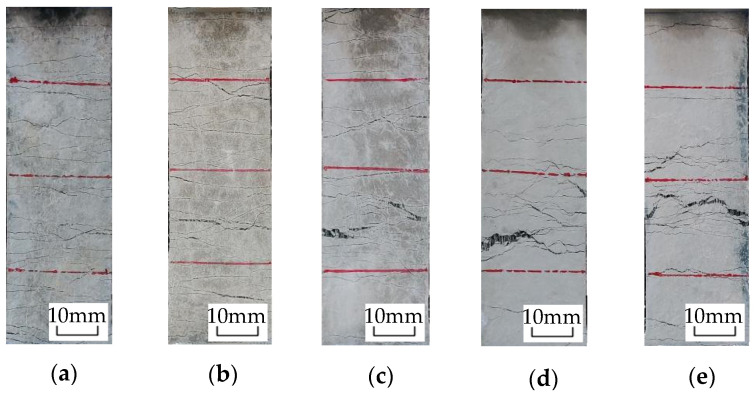
Failure mode of uniaxial tensile specimen: (**a**) Group A; (**b**) Group B; (**c**) Group C; (**d**) Group D; (**e**) Group E.

**Figure 11 materials-16-00697-f011:**
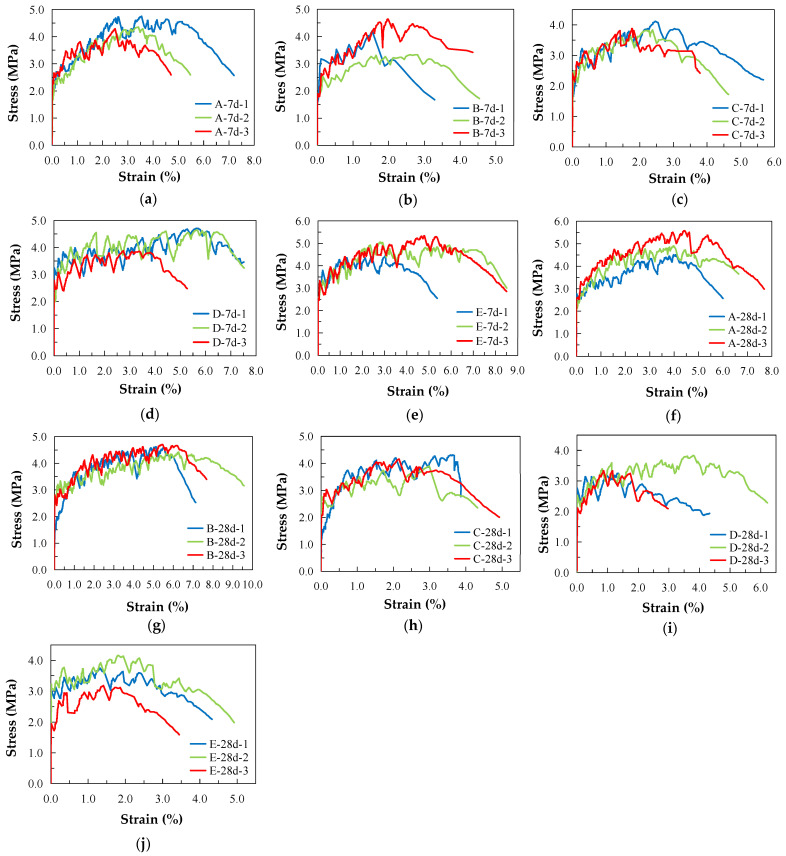
Uniaxial tensile stress–strain curve of ECCs with different sand sizes and ages: (**a**) Group A-7d; (**b**) Group B-7d; (**c**) Group C-7d; (**d**) Group D-7d; (**e**) Group E-7d; (**f**) Group A-28d; (**g**) Group B-28d; (**h**) Group C-28d; (**i**) Group D-28d; (**j**) Group E-28d.

**Figure 12 materials-16-00697-f012:**
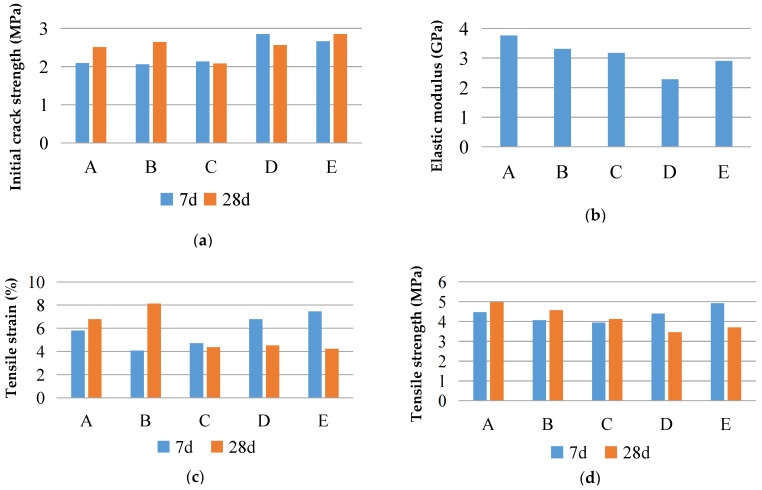
Uniaxial tensile properties of specimens with different particle sizes and ages: (**a**) initial crack strength; (**b**) elastic modulus; (**c**) tensile strain; (**d**) tensile strength.

**Figure 13 materials-16-00697-f013:**
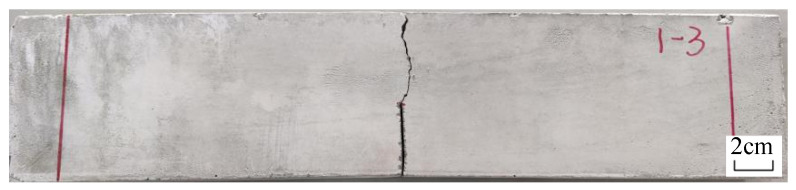
Failure mode of the three-point bending test specimen.

**Figure 14 materials-16-00697-f014:**
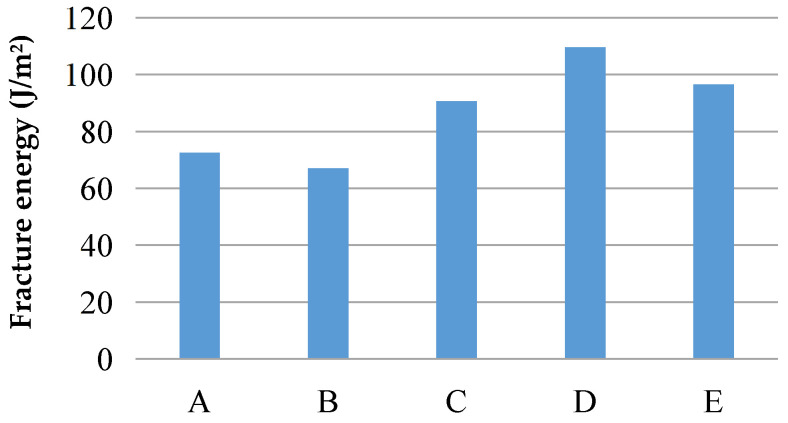
Fracture energy of the cement matrix with different sand particle sizes.

**Figure 15 materials-16-00697-f015:**
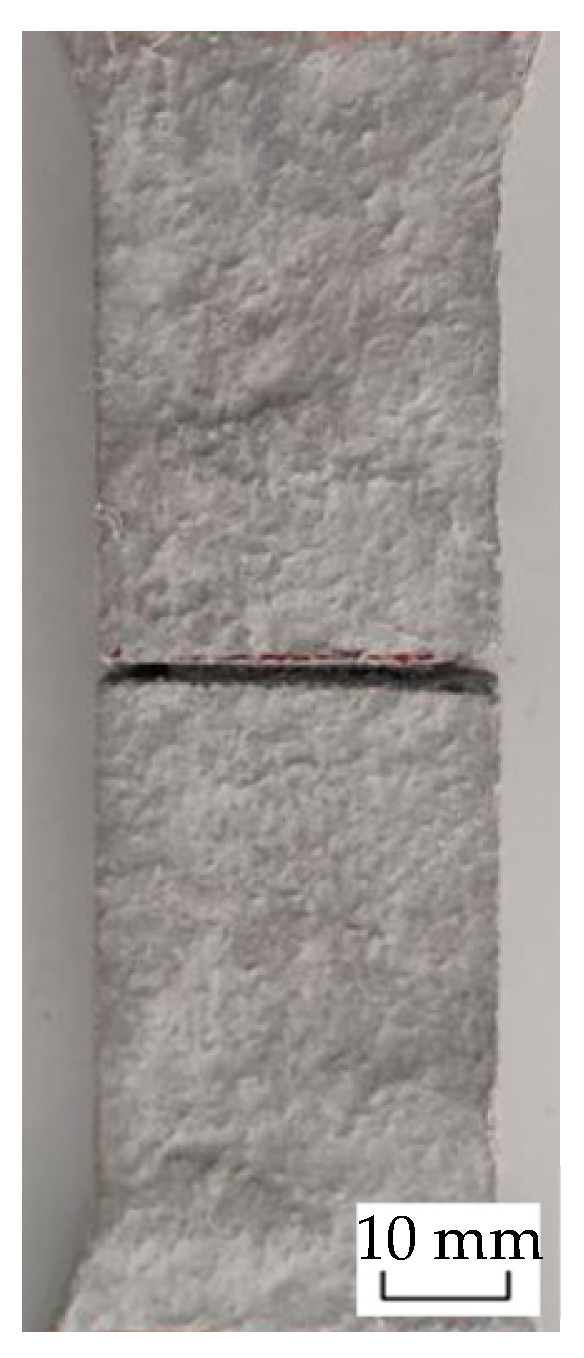
Failure mode of the single-seam tensile specimen.

**Figure 16 materials-16-00697-f016:**
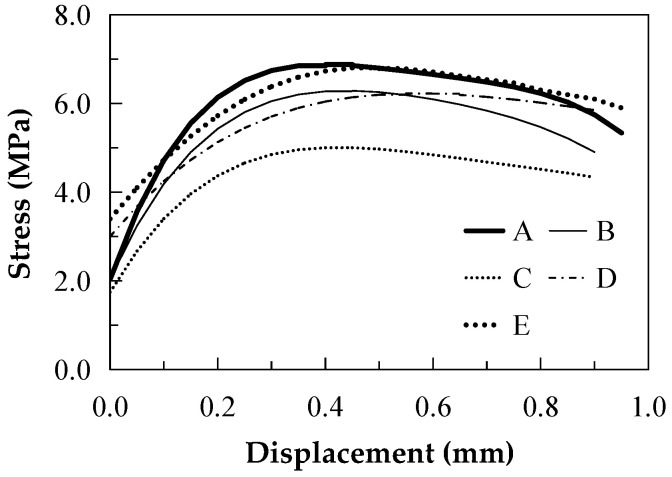
Tensile stress–displacement curve of the single seam with different sand particle sizes.

**Figure 17 materials-16-00697-f017:**
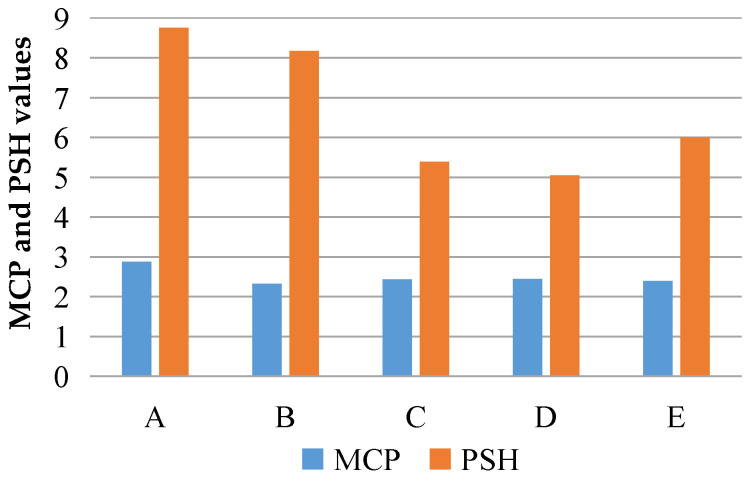
Comparative analysis of MCP and PSH values for the desert sand and ordinary sand.

**Table 1 materials-16-00697-t001:** Chemical composition of desert sand.

Component	SiO_2_	Al_2_O_3_	Fe_2_O_3_	CaO	MgO	TiO_2_	MnO	SO_3_
Proportion (%)	74.48	9.53	2.78	4.34	1.76	0.34	0.001	0.08
Component	Cl	MnO	P_2_O_5_	Cl^−1^	Na^+1^	K^+1^	Alkali content	Loss on ignition
Proportion (%)	0.013	0.001	0.09	0.005	0.007	0.003	2.67	3.34

**Table 2 materials-16-00697-t002:** Physical and mechanical properties of PE fiber.

Diameter (μm)	Length (mm)	Tensile Strength (MPa)	Elastic Modulus (GPa)	Elongation (%)	Density (g/cm^3^)
24	12	3000	120	5	0.97

**Table 3 materials-16-00697-t003:** ECC mixing proportions (kg/m^3^).

Cement	Slag	Sand	Water	Fiber	Water Reducer	Thickener
617.2	411.4	308.6	250	12	6	0.5

**Table 4 materials-16-00697-t004:** Fracture energy of the ECC matrix with different sand sizes.

Group	Peak Load *F* (kN)	Fracture Toughness *K*_m_ (MPa·m^1/2^)	Fracture Energy *J*_tip_ (J/m^2^)
A	0.71	0.522	72.5
B	0.64	0.471	67.0
C	0.73	0.536	90.6
D	0.68	0.500	109.6
E	0.72	0.529	96.5

**Table 5 materials-16-00697-t005:** Complementary energy of the ECC single-seam cracking test.

Group	Peak Stress *σ*_0_ (MPa)	Opening Width Corresponding to Peak Stress *δ*_0_ (mm)	Complementary Energy *J*′_b_ (J/m^2^)
A	7.24	0.40	635
B	6.14	0.48	547.5
C	5.07	0.45	397.7
D	6.27	0.63	553.9
E	6.83	0.52	578.0

**Table 6 materials-16-00697-t006:** MCP and PSH values of the desert sand and ordinary sand with different particle sizes.

Group	MCP	PSH
A	2.88	8.76
B	2.33	8.17
C	2.44	5.39
D	2.45	5.05
E	2.40	6.00

## Data Availability

The data used to support the findings of this study are included within the article.

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
