# Peer review of "Experimental Study on Toughness of Engineered Cementitious Composites with Desert Sand"

_materials, 2023, doi:10.3390/ma16020697_

Round 1

Reviewer 1 Report

The article addresses an important and very interesting topic of the experimental study on toughness of engineered cementitious composites with desert sand, which is appreciated. The study includes the experimental research. In this paper, the engineered cementitious composites (ECC) with desert sand instead of ordinary sand were prepared, and the toughness properties of engineered cementitious composites (ECC) were studied. The Reviewer has some concerns regarding the whole paper (introduction, description of research, results, conclusions and references), because this paper was carelessly prepared. Generally, in this paper the English language should be improved. Some sentences were not clear and please use passive voice. In opinion of Reviewer this paper should be subjected to major revision.

Other comments:

1.          Generally, this paper is hard to understand – language and style!

2.          Introduction – what is the aim of this research? What is novelty of this research?

3.          Figures – the Reviewer cannot find figure 5 and figure 7. In addition, the figure 9 are not clear. All line is the same, thus the Reviewer cannot understand the results. Please improve it.

4.          Discussion – the Reviewer cannot see discussion of results and comparison these results with other papers/results. Authors describes other results from research (in introduction), also comparison these results to own results are necessary. Thus, please improve discussion of results.

5.          Conclusion - this is end of your research? Please improved this point. In addition, these conclusions are really poor. The Reviewer cannot see the most important conclusions from this research.

·       References – Authors cited many papers but the style and preparation of this paper is very poor (unacceptable). Below you can find few papers with the description of research of paper the level were satisfactory:

·       https://doi.org/10.3390/ma15228259

·       https://doi.org/10.3390/ma12162602

·       https://doi.org/10.3390/ma12162577

Finally, I hope that my comments will be helpful for the authors.

Reviewer 2 Report

1. The results presented in the abstract should be quantitative.

2. In the introduction section, you provided 11 references for one short phrase. This is not interest. Please remove at least 8 items or present them separately in other sections.

3. State the innovation essay at the end of section 1. Also, atargeted and written flowchart of the steps of the article is also provided.

4. Add the following references to the text:

* Assessment of post-heat behavior of cement mortar incorporating silica fume and granulated blast-furnace slag,Journal of Structural Fire Engineering 11 (2), 221-246.

* Strength of SCLC Recycle Springs and Fibers Concrete Subject to High Temperatures، Structural Concrete.

* Cementitious Mortars containing Pozzolans under Elevated Temperatures, Structural Concrete.

5. How was the information in Table 1 obtained?

6. What standard are the materials presented in part 2.2 and other parst based on? Please provide.

7. Provide the unit of measurement for the data in Table 3.2

Round 2

Reviewer 1 Report

Thank you for your improving. In my opinion you should use the passive voice in the case of description of research, results, discussions and conclusions (please check by Native speakers). In addition, please use bullets in conclusions, and please shown the most important conclusions. 

Finally, I hope that my comments were helpful for Authors.

Reviewer 2 Report

accept

Author Response

Thank you very much for your comments and recognition.